# Novel CDK12/13 Inhibitors AU-15506 and AU-16770 Are Potent Anti-Cancer Agents in EGFR Mutant Lung Adenocarcinoma with and without Osimertinib Resistance

**DOI:** 10.3390/cancers15082263

**Published:** 2023-04-12

**Authors:** Tapan K. Maity, Eun Young Kim, Constance M. Cultraro, Abhilash Venugopalan, Leena Khare, Ramulu Poddutoori, Sivapriya Marappan, Samiulla D. Syed, William G. Telford, Susanta Samajdar, Murali Ramachandra, Udayan Guha

**Affiliations:** 1Thoracic and GI Malignancies Branch, CCR, NCI, NIH, Bethesda, MD 20892, USA; 2Aurigene Discovery Technologies Ltd., Bangalore 560100, India; 3Experimental Transplantation & Immunotherapy Branch, CCR, NCI, NIH, Bethesda, MD 20892, USA

**Keywords:** lung cancer, EGFR, osimertinib, resistance, AU15506, AU16770, CDK12, CDK13, inhibitor

## Abstract

**Simple Summary:**

Potential application of CDK12/13 inhibitor in overcoming resistance to osimertinib, a third-generation EGFR TKI, in vitro and in vivo.

**Abstract:**

Osimertinib is a third-generation epidermal growth factor receptor and tyrosine kinase inhibitor (EGFR-TKI) approved for the treatment of lung adenocarcinoma patients harboring EGFR mutations. However, acquired resistance to this targeted therapy is inevitable, leading to disease relapse within a few years. Therefore, understanding the molecular mechanisms of osimertinib resistance and identifying novel targets to overcome such resistance are unmet needs of cancer patients. Here, we investigated the efficacy of two novel CDK12/13 inhibitors, AU-15506 and AU-16770, in osimertinib-resistant EGFR mutant lung adenocarcinoma cells in culture and xenograft models in vivo. We demonstrate that these drugs, either alone or in combination with osimertinib, are potent inhibitors of osimertinib-resistant as well as -sensitive lung adenocarcinoma cells in culture. Interestingly, only the CDK12/13 inhibitor in combination with osimertinib, although not as monotherapy, suppresses the growth of resistant tumors in xenograft models in vivo. Taken together, the results of this study suggest that inhibition of CDK12/13 in combination with osimertinib has the potential to overcome osimertinib resistance in EGFR mutant lung adenocarcinoma patients.

## 1. Introduction

Lung cancer is the leading cause of cancer-related deaths among both men and women in the United States with 5-year survival rate of just 19%. Advances in genetic profiling of lung adenocarcinoma, the most common subtype of non-small-cell lung carcinoma, has led to the identification of several oncogenic drivers. Most prominent among them are *EGFR*, *KRAS*, and *MET* [1]. Specific mutations in the EGFR tyrosine kinase domain sensitize lung adenocarcinoma tumors to respond differentially to EGFR tyrosine kinase inhibitors (TKIs). Osimertinib, a third-generation EGFR TKI, targets mutant EGFR harboring both TKI-sensitizing and first/second-generation EGFR TKI-resistant T790M mutation in lung adenocarcinoma patients [2,3]. Unfortunately, all patients treated with osimertinib eventually develop acquired resistance. Recent studies have revealed various mechanisms of acquired resistance to osimertinib [4,5,6,7,8,9,10]. However, no resistance mechanism can be identified in a significant number of patients. Most importantly, a large number of identified resistance mechanisms do not reveal an actionable target. Therefore, further studies are necessary to develop new therapies that are agnostic to specific resistance mechanisms; therapies that can target resistant tumors with either an unidentified resistance mechanism or with an identified, hitherto unactionable target of resistance. It is worth exploring the pathways that are not apparently linked to the EGFR pathway or compensatory parallel signaling pathways for targeted therapy. This may include pathways essential for cell survival and resistance in tumors, such as DNA repair or the cell cycle. Inhibition of DNA repair mechanisms or the cell cycle may be exploited to provide an additive or synergistic effect to osimertinib treatment of EGFR mutant tumors.

The cell cycle plays an important role in mammalian cell proliferation [11]. The cell cycle can be regulated either by cyclin-dependent kinases (CDKs, catalytic subunit) in association with cyclins (regulatory subunit) or by cell-cycle checkpoint sensors to correct deficiency in replication or chromosome separations during cell divisions [12,13]. Cell-cycle checkpoint inhibitors that target cell-cycle components, specifically CDKs, could serve as potential therapeutic agents to treat human cancer [12]. CDK1-6 bind multiple cyclins and are directly involved in cell-cycle progression, whereas CDK7-20 bind to single cyclins and are involved in transcriptional regulation [12,14]. CDK7, CDK12, and CDK13 regulate transcription by phosphorylating the C-terminal domain (CTD) of RNA polymerase II. CDK7 is involved in phosphorylation of serine-5 and serine-7 residues of RNA polymerase II CTD to initiate transcription, whereas CDK12 and 13 are involved in phosphorylation of serine-2 residue to promote elongation of the RNA polymerase II transcription complex. CDK12 and CDK13 inhibitors are known to inhibit transcription of DNA repair genes leading to inhibition of DNA repair mechanisms and causing cell death [15]. CDK12 function and its role as a target for cancer treatment has been reviewed recently [16]. We have also shown that non-functional G879V mutation in CDK12 results in reduced expression of long transcript genes that include DNA repair pathway genes, demonstrating that this mutation affects CDK12 DNA damage repair function [17,18]. We hypothesized that CDK12/13 inhibition will be able to circumvent both known and unknown mechanisms of resistance to osimertinib by virtue of targeting the DNA repair pathway that is independent of the hitherto known mechanisms of resistance to osimertinib. In this study, we sought to examine the role of CDK12/13 inhibitors in overcoming osimertinib resistance in human tumors. First, we developed osimertinib-resistant cells by long-term culture of sensitive cells in the presence of osimertinib to replicate in vivo selection of resistant tumors in patients treated with osimertinib. These resistant human lung adenocarcinoma cell lines were tested for growth inhibition in vitro by two novel CDK12/13 inhibitors. We further assessed the potency of these drugs in an osimertinib-resistant mouse xenograft model and demonstrated that these CDK12/13 inhibitors, in combination with osimertinib, can overcome osimertinib resistance in vivo. These results raise the possibility of the evaluation of CDK12/13 inhibitors in clinical studies in combination with osimertinib to overcome resistance.

## 2. Results

### 2.1. Establishment of Osimertinib-Resistant Lung Adenocarcinoma Cell Lines

To gain a better understanding of the mechanisms underlying osimertinib drug resistance and to develop more effective therapeutic options for lung adenocarcinoma, we generated and characterized several osimertinib-resistant cell line models of acquired drug resistance (Figure 1). EGFR mutant lung adenocarcinoma cell lines H1975 (L858R/T790M), PC9 (Del. E746–A750), and HCC827 (Del. E746–A750) were treated for three months with increasing doses (5 nM–2000 nM) of osimertinib to generate osimertinib-resistant cell lines (Figure 1A). We verified drug resistance by performing cell-growth-inhibition assays (IC_50_, half maximal inhibitory concentration) on parental and resistant cells in the presence of osimertinib. While the parental cells are sensitive to osimertinib with IC_50_ ranging from 7.5 nM for HCC827 cells to 46.8 nM for H1975 cells, the isogenic resistant cell lines had significantly lower sensitivity with IC_50_, ranging from 1.86 µM for H1975-osiR cells to 5.49 µM for HCC827-osiR cells (Figure 1B–D). Next, we compared the growth properties of H1975 and PC9 parental and resistant cells by performing cell growth assays in a drug-free medium. Interestingly, resistant cells grew slower than their corresponding parental cells, with a profound growth reduction (~50%), as observed in PC9-osiR cells (Figure 1E). To examine whether EGFR downstream signaling is affected in resistant cells, we performed Western blot analyses for MAPK and PI3K signaling pathway components in H1975/H1975-osiR and PC9/PC9-osiR cells grown in drug-free conditions or in medium containing 50 nM osimertinib (Figure 1F). EGFR levels were slightly higher in resistant cells, but phospho-EGFR (pEGFR) levels were significantly lower in untreated resistant cells, which reduced further upon osimertinib treatment. To dissect downstream proliferation, survival, and metabolic pathways affected in the resistant cell lines, we examined the phosphorylation profiles of AKT and ERK.

We observed that osimertinib treatment of sensitive cells reduced pAKT and pERK as expected, but there was less inhibition of pAKT and pERK in resistant cells. Likewise, we found that pERK (T202/Y204) basal levels were lower in osimertinib-resistant cells and remained largely unaffected upon osimertinib treatment. Similarly, osimertinib treatment decreased S6 phosphorylation of parental but not resistant cells. These results indicate that although osimertinib is effective in blocking the function of activated EGFR in resistant cells, the downstream signaling components are still capable of inducing cell survival and proliferation signals. Because AKT and ERK lie downstream of a complex network of receptor signaling pathways, other survival or proliferative pathways may be activated in osimertinib-resistant cells.

To further examine if osimertinib resistance in the above cell lines developed due to known mechanisms of osimertinib resistance, such as C797S mutation or overexpression of MET and HER2 [10], we performed exome sequencing and copy number assays on DNA isolated from sensitive and resistant cells. Interestingly, we did not detect either C797S mutation or any other mutation in common oncogenes. In addition, we also did not observe any amplification of *MET* and *ERBB2* (Appendix A). This indicates that some uncommon mechanism might be involved in conferring resistance in these cell lines. We used these cell line models to interrogate the efficacy of treatments that would be agnostic of the common mechanisms of osimertinib resistance and hence would have the potential to overcome resistance regardless of the common resistance mechanisms.

### 2.2. Characterization of Compounds for CDK12/13 Inhibition

Availability of a novel CDK12 inhibitor AU-15506 (Aurigene Discovery Technologies Ltd.) allowed us to test if this compound can inhibit receptor activity or intracellular kinases and therefore has potential anti-cancer activity. Subsequently, we used AU-16770, a more selective newer version of AU-15506. We performed a kinase profiling assay with a panel of kinases in the presence of 1 μM and 10 μM of these drugs and observed that both AU-15506 and AU-16770 inhibited few kinases, including PKC-alpha kinase; SGK1, a kinase downstream of the PI3K signaling pathway; and FGFR1 at approximately <50% inhibition (Appendix A). Hence, these compounds have off-target effects on some kinases in this in vitro kinase assay, but only at a high micromolar concentration.

To further characterize these inhibitors for their CDK activity, we evaluated AU-15506 and AU-16770 in CDK12 and CDK13 biochemical assays (TR-FRET) and identified them as sub-nanomolar CDK12 and CDK13 inhibitors (Figure 2A). These inhibitors were also able to inhibit CDK7 and CDK9, although at higher concentrations. We also performed a cell viability profile of Jurkat cells in the presence of AU-15506 and AU-16770, and the potencies of growth inhibition as measured by IC_50_ were found to be 35.3 nM and 37.6 nM, respectively (Figure 2B). This indicates that these compounds are very effective in inhibiting cell growth. To further test if these compounds effectively block CDKs in vitro, we performed a target engagement (CDK binding) assay in Jurkat cells treated with increasing concentrations of the drugs. We observed that IC50s of AU-15506 and AU-16770 for CDK12 occupancy were 9.7 nM and 12.0 nM, respectively, suggesting that these inhibitors bind CDK12 very effectively in the cells (Figure 2C). CDK12 phosphorylates Ser 2 (pS2) residue of RNA polymerase II. Hence, we examined if these inhibitors prevent phosphorylation of S2 of RNA Pol II in Jurkat cells. As expected, AU-15506 and AU-16770 were able to effectively inhibit phosphorylation at 121.2 nM and 15.33 nM concentrations, respectively (Figure 2D), further validating that these inhibitors can effectively block CDK12 substrate phosphorylation in vitro. Next, we performed pharmacokinetics profiling of AU-15506 and AU-16770 in CD-1 mice to examine the drugs’ absorption, bioavailability, distribution, metabolism, and excretion (Figure 2E). We observed that when the drugs were administered through an intravenous (1 mg/kg) route, the initial plasma concentrations (C_0_) appeared to be 1.453 μg/mL and 2.686 μg/mL for AU-15506 and AU-16770, respectively. Other pharmacokinetic parameters included the mean terminal elimination half-life (T_1/2_) which was 0.21 h and 0.2 h, respectively, and clearance (CL) which was 56 mL/min/kg and 48 mL/min/kg, respectively. The areas under plasma concentration curve (AUCs) were 294 ng·h/mL and 349 ng·h/mL, respectively. However, when the drugs were administered by oral gavage (PO; 10 mg/kg), the maximum plasma concentrations (C_max_) were found to be 1095 ng/mL and 1076 ng/mL for AU-15506 and AU-16770, respectively. Furthermore, the AUCs were 1785 ng·h/mL and 979 ng·h/mL, respectively. The percentages of the active drug that reached systemic circulation (bioavailability; %F) were 61% and 26% for AU-15506 and AU-16770, respectively.

### 2.3. CDK12/13 Inhibitors, AU-15506 and AU-16770, Inhibit Growth of Both Osimertinib-Sensitive and -Resistant Cells

CDK12/13 inhibitors have been demonstrated to inhibit cancer cell growth [19,20]. To test the ability of AU-15506 to inhibit the growth of osimertinib-resistant cells alone or in combination with osimertinib, we performed cell-growth-inhibition assays in H1975, PC9, and HCC827 parental cells, as well as their isogenic osimertinib-resistant counterparts (Figure 3A–C). After three days in culture, the inhibitor concentrations required to inhibit 50% cell growth (IC_50_) were calculated and summarized (Figure 3D). We also cultured these cells in specially designed tissue culture plates to generate three-dimensional (3D) spheres to mimic in vivo tumor organization and performed cell-growth-inhibition assays with AU-15506 alone or in combination with osimertinib (Figure 3B). Although in HCC827-osiR-NCI1 cells the IC_50_ of AU-15506 and osimertinib combination (72.8 nM) appeared to be higher than AU-15506 alone (38.6 nM), it was either similar or lower in PC9-osiR-NCI1 or H1975-osiR-NCI1 cell lines compared to their sensitive counterparts. Therefore, our results clearly show that AU-15506, either alone or in combination with osimertinib, is very effective (IC_50_ < 75 nM) in preventing growth of H1975, PC9, and HCC827 parental and osimertinib-resistant cells both in two-dimensional (2D) and 3D cell growth. Similarly, cell growth inhibition by AU-16770, the second most selective CDK12 inhibitor, was examined for its ability to inhibit cell growth both in parental and osimertinib-resistant H1975, PC9, and HCC827 cells (Figure 3C,D). Our three-day cell-growth-inhibition end-point assay with AU-16770 either as a single agent or in combination with osimertinib resulted in IC_50_ of AU-16770 between 5.8 nM to 40.3 nM, demonstrating the effectiveness of AU-16770 in inhibiting the growth of both osimertinib-sensitive and-resistant cells.

### 2.4. AU-15506 or AU-16770 in Combination with Osimertinib Exhibit Anti-Tumor Activity and Increases Overall Survival in A Mouse Xenograft Model

To interrogate the anti-tumor efficacy of AU-15506 or AU-16770 in vivo, we first generated a mouse xenograft model by subcutaneous injection of either PC9-osiR-NCI1 (Figure 4A–C, Appendix A) or H1975-osiR-NCI1 (Appendix A) cells into athymic female nude mice. Initially, these mice were treated with osimertinib (5 mg/kg/day) to maintain osimertinib resistance. After 2 weeks, the mice were randomized and administered vehicle (0.5% methyl cellulose), osimertinib (5 mg/kg/day), AU-15506 (5 mg/kg/day), or AU-16770 (5 mg/kg/day), either as a monotherapy or in combination with osimertinib. Similar to the vehicle treatment, AU-15506 as a single agent failed to inhibit tumor growth in the xenograft model (Appendix A). Osimertinib monotherapy induced some tumor regression, suggesting residual sensitivity of these resistant cells in vivo. However, when AU-15506 was administered in combination with osimertinib, significant tumor shrinkage was observed within 14 days (Appendix A). Similarly, AU-16770 was also found to be very effective in preventing xenograft tumor growth at 28 days when administered in combination with osimertinib (Figure 4A–E). We also measured tumor growth inhibition (TGI) of these mice at 28 days, which was measured by (1 − mean value of tumor group/tumor volume of control group) × 100 [21]. We observed that TGI reached 50% upon osimertinib treatment and 71.5% upon treatment with osimertinib in combination with AU-16770 (Figure 4B,C). In addition, we found a significant difference in tumor growth inhibition between osimertinib alone and combination treatment using osimertinib and AU-16770 (days 0–32) or osimertinib and AU-15506 (days 33–46) after 39 days of continuous dosing. Finally, Kaplan–Meier survival curves indicated that the median survival of the vehicle-treated mice (~29 days) was lower than the osimertinib-treated mice (40 days). However, when mice were treated with AU-16770 and osimertinib combination therapy, 85% of the mice survived 46 days post-treatment and medial survival was unreached (Figure 4E). These results indicate that this combination therapy is quite effective, warranting further evaluation of CDK12/13 inhibitor use in overcoming osimertinib resistance in tumors.

### 2.5. Cell-Cycle Alterations upon AU-16770 Treatment of PC9 and PC9-osiR-NCI1 Cells

As a first step toward gaining an understanding of the molecular mechanism by which CDK12/13 inhibitors inhibit osimertinib-resistant tumor growth, we investigated the effect of these drugs on cell-cycle progression. Because CDK12/13 reported to phosphorylate serine 2 within the C-terminus of RNA Pol II [13,22], we first determined the optimum concentration of AU-16770 needed to inhibit this effect. Western immunoblot analyses revealed that a range of 100 nM–1 uM AU-16770 inhibited RNA Pol II-serine 2 phosphorylation (Figure 5A). Next, we performed cell-cycle analyses of PC9 and PC9-osiR-NCI1 cells treated with vehicle (DMSO), osimertinib (50 nM), or AU-16770 (100 nM) for 18 h. Cell-cycle profiles (Figure 5B) were used to quantify the proportion of cells distributed in G1, S, and G2 (Figure 5C). A majority of vehicle-treated PC9 parental cells were distributed in G1 (43.3%) and S (49.9%), while a very low percentage was distributed in G2 (4.55%). Likewise, a majority of vehicle-treated PC9-osiR-NCI1 were distributed in G1 (53.9%) and S (40.8%), while a low percentage was distributed in G2 (1.98%). However, treatment of PC9 parental cells with either osimertinib or AU-16770 caused G2 phase accumulation (24.8% and 30.6%, respectively), suggesting these inhibitors block the G2-M transition of these cells. In contrast, osimertinib treatment of PC9-osiR-NCI1 cells did not change this profile, while AU-16770 treatment resulted in G2 accumulation (33.5%). Taken together, AU-16770 blocked the G2-M transition of osimertinib-resistant cells, similar to the sensitive parental cells (Figure 5C). These results indicate that AU-16770, a cell-cycle check point inhibitor, blocks cell-cycle progression of osimertinib-resistant PC9-osiR-NCI1 cells, lending further credence to its use in treating osimertinib-resistant lung adenocarcinoma.

### 2.6. Identification of Cellular Signaling Components That Are Affected by AU-16770

In order to understand how CDK12/13 inhibitors such as AU-16770 affect the phosphorylation of EGFR downstream signaling components and overcome osimertinib resistance, we treated H1975, H1975-osiR-NCI1, PC9, and PC9-osiR-NCI1 cells with either DMSO (vehicle) or AU-16770 and performed immunoblot analyses of lysates prepared from these cells after three and six hours of treatment (Figure 6A). We found that CDK12 levels were not affected by this drug treatment. Although total EGFR levels were slightly higher, pEGFR levels were significantly lower in both H1975-osiR and PC9-osiR-NCI1 cells compared to their parental counterparts. However, neither the levels of total EGFR nor pEGFR, in both parental and resistant cells, were affected by AU-16770 treatment. Phosphorylation of serine 2 within the CTD of RNA Pol II is a hallmark of CDK12 kinase activity. Within three hours of AU-16770 treatment, we found a reduction in RNA Pol II serine 2 phosphorylation in both parental and resistant H1975 and PC9 cells. Next, we examined if phosphorylation and expression of ATR, a protein kinase involved in DNA damage repair, is affected upon AU-16770 treatment of osimertinib-resistant cells. However, we did not observe any effect on ATR expression or phosphorylation. There is a recent report which states that CDK12 may be involved in EGF-mediated optimal transcription of cFOS proto-oncogene [22]. In addition, another study also reported that targeted therapy induces transcription of a specific set of genes, including cFOS in a subpopulation of cells that become drug tolerant [23]. THZ1, a CDK7/CDK12 inhibitor, prevents the emergence of drug resistance by remodeling enhancers and inhibiting dynamic transcription of these genes [23]. Hence, we examined whether expression and phosphorylation of cFOS is affected by AU-16770 treatment. We observed that compared to osimertinib-sensitive cells, basal level of cFOS is reduced in resistant cells. However, upon AU-16770 treatment, both the expression and phosphorylation of cFOS were significantly increased.

In order to examine whether expressions of long transcript genes were affected, we analyzed expression of DNA repair genes along with EGFR signaling pathway genes following treatment of PC9-osiR cells with DMSO (vehicle) and 15 nM (IC50) of AU-16770 for 8 h, 24 h, and 48 h. We performed quantitative PCR and Western blot analyses of mRNAs and whole-cell extracts isolated from treated cells. Our results indicate that mRNA expression of *ATM*, *ATR*, *BRCA1*, *BRCA2*, *FANC1*, *MDC1*, *NEK9*, *ORC3*, *RAD51D*, and *TERF2* were significantly reduced at 48 h of treatment (Figure 6B). In contrast, the reduction in *CHK1* was not significant upon AU-16770 treatment. Western blot analyses revealed that although there was no significant inhibition of RNA Pol II serine 2 phosphorylation at the low 15 nM dose of AU-16770, the phosphorylation of AKT (T308, S473) that is required for cell survival was reduced upon AU-16770 treatment at 24 and 48 h (Figure 6C). Expression of ATR, ATM, and RAD51 protein reduced at 24–48 h upon AU-16770 treatment (Figure 6D). On the other hand, there was no significant change in protein expression of CHK1 and CHK2 (Appendix A), similar to the unchanged mRNA expression (Figure 6B), indicating similar trends in mRNA and protein expression levels.

Our data indicate that while osimertinib inhibits the downstream RAS-MAPK and PI3K-AKT pathways by targeting mutant EGFRs, a CDK12/13 inhibitor induces vulnerabilities in DNA repair and cell-cycle progression, thereby targeting parallel non-overlapping pathways to circumvent EGFR TKI resistance in combination with osimertinib (Figure 6E).

## 3. Discussion

In this study, we leveraged the availability of CDK12/13 inhibitor AU-15606 and AU-16770 to test if these compounds could overcome osimertinib resistance in lung adenocarcinoma cells. CDK12 plays an important role as a regulator of the expression of DDR genes that happen to have long transcripts. A recent study showed that loss of CDK12 in cancer cells results in premature cleavage and polyadenylation of DDR gene transcripts, leading to decreased DNA damage response and lower recovery of affected cells [15]. We anticipated similar outcomes in response to AU-15606 and AU-16770 CDK12/13 inhibitors.

Initially, we generated isogenic drug-resistant EGFR mutant lung adenocarcinoma cell lines by osimertinib dose escalation. We utilized this model system to study the mechanism of osimertinib resistance with the intention of developing more effective anti-cancer therapies. It is well documented that cancer cells develop drug resistance due to several mechanisms, including tumor heterogeneity, EMT, and inhibition of cell death [24]. Here, we showed that these osimertinib-resistant cells (H1975-osiR-NCI1, PC9-osiR-NCI1, and HCC827-osiR-NCI1/2) grow slower than the parental cells, suggesting that altered proliferation may be associated with drug resistance (Figure 1E). We further demonstrated that both the AKT survival and MAPK (ERK) cell proliferation pathways remain active even after osimertinib treatment of resistant cells (Figure 1F). Interestingly, the basal pERK level was lower in the osimertinib-resistant cells compared to the sensitive cells. This is related to the significantly lower level of pEGFR in the resistant cells (Figure 1F, pEGFR, lanes 2 and 4) and phenotypically correlates with the reduced growth observed in the resistant cells compared to the sensitive cells (Figure 1E). However, while osimertinib treatment of parental H1975 and PC9 cells almost completely abrogates pERK levels, there is incomplete inhibition of ERK phosphorylation in the osimertinib-resistant cells, suggesting that residual pERK levels in the presence of osimertinib continue to promote growth signals and hence resistance to osimertinib (Figure 1F, pERK lanes 6 and 8). This provides the resistant cells with a growth advantage compared to parental cells.

Under normal cell growth conditions, pERK translocates to the nucleus and activates transcription factors such as FOS, ELK, and other proteins resulting in cell-proliferation-specific gene expression. Unphosphorylated ERK is retained in the cytoplasm and cannot activate the above genes. Furthermore, cytoplasmic ERK activates proapoptotic genes, such as *DAPK,* which inhibits growth and promotes apoptosis [25]. It is still possible that a parallel receptor tyrosine kinase pathway is activated in these resistant cells. We have previously demonstrated that pAKT is activated in PC9-osiR and HCC827-osiR cells harboring EGFR Del746-750 mutant. Furthermore, dactolisib, a dual PI3K/AKT inhibitor, inhibits activation of AKT and circumvents osimertinib resistance [26].

In order to target pathways other than EGFR signaling to develop successful combination therapies for overcoming osimertinib resistance, we assessed the potency of two novel CDK12/13 inhibitors, AU-15506 and AU-16770. Initially, we characterized these lead compounds for their effectiveness in inhibiting CDKs, particularly CDK12, CDK13, CDK7, and CDK9, in an in vitro kinase assay. We observed that these compounds inhibit CDK12 and 13 more efficiently than CDK7 and CDK9 (Figure 2A). These compounds were found to have a minimal off-target effect at micromolar concentrations when tested in a panel of 27 other kinases (Appendix A). In our study, the compounds AU-16770 and AU-15506 tested at 1 μM and 10 μM showed high selectivity in inhibiting CDK12/13, while some marginal activity was observed for SGK1, GLK, and FGFR1 kinases at micromolar concentrations of the drugs (Appendix A). High selectivity is expected to increase the therapeutic index of these inhibitors. There are not well explored correlations for activities of SGK1 and GLK with CDK12/13. However, CDK12 inhibition may downregulate the expression of FGFR1 and other FGF receptors, potentially inhibiting the signaling pathways that promote cancer cell growth and survival. The CDK12 inhibitor reduced the expression of FGFR1 and other FGF receptors in ovarian cancer cells, leading to decreased cell proliferation and increased apoptosis. The reference molecule CDK12/13 inhibitor, THZ531 [27], also showed 82% inhibition of FGFR1 at 1 μM concentration of the drug. However, AU-16770 and AU-15506 inhibits <50% at 1 μM and this inhibition may not result in additional toxicities.

Next, in the Jurkat cell viability assay, these compounds showed high inhibitory activity as determined by their very low IC_50_ (Figure 2B). When we performed target engagement assay in Jurkat cells, we observed that these compounds bind irreversibly to the intended target, CDK12, more efficiently than CDK7, another member of CDK family with high kinase domain similarity (Figure 2C). We further validated the effectiveness of these compounds by examining inhibition of phospho-serine modification of RNA pol II, a substrate of CDK12 (Figure 2D). Therefore, we concluded that AU15606 and AU-16770 are CDK12/13-specific highly potent anti-cancer agents in vitro. Finally, we performed pharmacokinetic assays of AU-15506 and AU-16770 in mice and analyzed the pharmacokinetic parameters. When administered through intravenous injection, these compounds reached the highest concentration in the blood stream in ~0.2 h, and drug clearance appeared to be similar between the two compounds (56 mL/min/kg vs. 48 mL/min/kg). When these drugs were administered through oral gavage (PO), the bioavailability of these compounds appeared to vary significantly (61% vs. 28%). However, despite bioavailability nonequivalence, AU-16770 had better therapeutic equivalence. Furthermore, these compounds appeared to be more specific and effective (IC50s: 0.035 nM and 0.038 nM) in Jurkat cells compared to THZ531 (IC50 = 50 nM), a first-in-class covalent inhibitor of CDK12/13 [18]. Therefore, these compounds need to be validated further in vitro and in vivo as effective anti-cancer agents to inhibit osimertinib-resistant tumor cell growth.

We found that CDK12/13 inhibitors, AU-15506 and AU-16770, are very effective in inhibiting osimertinib-resistant H1975-osiR-NCI1, PC9-osiR-NCI1, and HCC827-osiR-NCI1/2 cancer cell growth in vitro, either alone or in combination with osimertinib (Figure 3A–C). However, AU16770 was only effective in reducing xenograft tumor growth when combined with osimertinib (Figure 4A–H). This suggests that osimertinib-resistant patients may still need to take osimertinib and other novel drugs to prevent disease recurrence and survive longer. Indeed, we demonstrated that a lung adenocarcinoma patient with *ERBB2* amplification and harboring CDK12-G879V mutation responded to HER2-directed targeted therapy in combination with standard chemotherapy that explored the vulnerability of impaired DNA damage repair caused by the CDK12-G879V mutation [18].

Proliferation of cells depend on normal duplication of genetic materials and cell divisions using the highly regulated cell-cycle process, and aberration may lead to cancer development. This process is controlled by CDKs (CDK4-6) in association with cyclins. Therefore, inhibitors that block CDK activities are potential anti-cancer agents [28]. Here, we demonstrated that the CDK12/13 inhibitor AU-16770 prevented the transition of osimertinib-resistant cells from the G2 to M phase, consistent with other studies (Figure 5B,C). It is interesting to note that because CDK12 is required during DNA transcription, depletion or inhibition of CDK12 activity is expected to arrest cells at the G1/S phase [29]. Further studies are necessary to understand how AU-16770 is responsible for growth arrest of PC9-osiR-NCI1 cells at the G2/M phase.

Phosphorylation and dephosphorylation of intracellular signaling components play an important role in modulating cell survival and cell proliferation. We performed immunoblot analyses of several intracellular proteins to determine if their phosphorylation status correlated with cellular AU-16770 sensitivity (Figure 6). To our surprise, we observed that AU-16770 treatment increases the synthesis of cFOS, leading to higher phospho cFOS proteins both in osimertinib-sensitive and -resistant cells (Figure 6A). Further experiments are clearly necessary to understand this effect. Quantitative PCR analysis of long transcripts associated with DNA damage response (DDR) genes indicated that, with the exception of *CHEK1*, expression of *ATR*, *ATM*, *BRCA1*, *BRCA2*, *FANC1*, *MDC1*, *NEK9*, *ORC3*, *RAD51*, and *TERF2* were significantly reduced in PC9-osiR cells upon treatment with AU-16770 for 48 h (Figure 6B), supporting impaired DDR and cell-cycle arrest. Previously, we have also shown that the G879V mutation in CDK12 affects its DNA repair function and makes cells susceptible to chemotherapy [18]. Interestingly, in our Western blot analyses (Figure 6C) of cell extracts of PC9-osiR-NCI cells treated with AU-16770, we did not demonstrate a reduction in RNA Pol II serine 2 phosphorylation as shown in Figure 5. This is presumably due to usage of a much lower (15 nM) concentration of AU-16770 compared to 100–300 nM that was used in the experiment shown in Figure 5. However, there was a reduction in protein expression of some DDR proteins, such as ATM, ATR, and RAD51 at 24–48 h upon AU-16770 treatment (Figure 6C,D).

Recently, it has been reported that inhibition of RAD51 is associated with decreased cervical cancer cell proliferation in vitro and in cervical cancer xenografts by attenuating cell-cycle transition [30]. Therefore, it is possible that osimertinib and AU-16770 combination treatment results in inhibition of RAD51 expression and ATR and cFOS phosphorylation, leading to the suppression of proliferation-specific gene expression and inhibition of cell-cycle transition of osimertinib-resistant cells. Based on our results, we have modeled the activation of different pathways in osimertinib-resistant cells and suppression of these pathways in the presence of AU-15506/AU-16770 (Figure 6E). We conclude that osimertinib and AU-16770 work synergistically to attenuate gene expression and protein synthesis of a specific set of genes that are critical for cell survival and cell proliferation [23,30].

In this study, we demonstrated that osimertinib-resistant cells generated in culture may serve as an excellent model system for osimertinib resistance in patients. This system allowed us to identify new anti-cancer agents to circumvent resistance. The role of CDK12/13 inhibitors as anti-cancer therapeutic agents is well known, but this is the first study to explore the use of CDK12/13 inhibitors in overcoming osimertinib resistance. Interestingly, the CDK12/13 inhibitors used in this study were equally effective in inhibiting the growth of the parental osimertinib-sensitive cells in culture as a monotherapy, underscoring the DDR and cell-cycle progression pathway inhibition that is parallel and non-overlapping to EGFR pathway inhibition by osimertinib.

## 4. Materials and Methods

### 4.1. Reagents and Antibodies

RPMI-1640 tissue culture medium was obtained from Millipore-Sigma (Burlington, MA, USA) and FBS was obtained from GeminiBio (West Sacramento, CA, USA). All chemicals were obtained from Millipore-Sigma, unless stated otherwise. Complete minitab protease inhibitor (#11836170001), and PhosStop phosphatase inhibitor (#49068450001) were obtained from Millipore-Sigma. Nitrocellulose Western transfer sandwich kit (#1704271) was obtained from Bio-Rad laboratories, (Hong Kong, China). Tyrosine kinase inhibitor osimertinib was obtained from ChemieTek (#CT-A9291) and CDK12 inhibitors AU-15506 (AU-CBB-15506) and AU-16770 (AU-CBB-16770) were provided by Aurigene Discovery Technologies Ltd. (Bengaluru, India). Mouse anti-EGFR mAb was obtained from BD Biosciences (#610017, East Rutherford. NJ, USA). Polyclonal or monoclonal antibodies to pEGFR-Y1068 (#2234), AKT (#4691), ERK (#4370), AMPK-alpha (#5831), ATR (#13934), ATM (#2873), RAD51 (#8875), CHK1 (#2360), CHK2 (#6334), pAKT (#4370, #4370), pERK (#4370), pACC (#11818), and pS6 (#4858) were obtained from Cell Signaling Technology (Danvers, MA, USA). Polyclonal or monoclonal antibodies to CDK12 (ABE1861), RNA Pol II [630849], pSer 2 RNA Pol II (MABE953), and pSer 5 RNA Pol II (MABE954) were obtained from Millipore-Sigma.

### 4.2. Cell Lines

Parental H1975 (#CRL-5908) and HCC827 (#CRL-5908) cell lines were purchased from the ATCC (Manassas, VA, USA), and the PC9 cell line was obtained from the Varmus Laboratory (MSKCC, New York, NY, USA). All human lung adenocarcinoma cells were maintained in RPMI supplemented with 10% FBS, 100 units/mL penicillin, and 100 μg/mL streptomycin. Jurkat cells procured from ATCC (#TIB-152) were cultured in RPMI with 10% FBS. Cells were authenticated by short tandem repeat (STR) profiling using the AmpFLSTR Identifiler kit (ThermoFisher, Waltham, MA, USA, #4322288) at the Protein Expression Laboratory (NCI, Frederick, Bethesda, MD, USA).

### 4.3. Generation of Osimertinib-Resistant Cell Lines

Initially, H1975, PC9, and HCC827 parental cells were seeded at low density (5 × 10^5^/10 mL) in 10 cm Petri dishes containing RPMI-1640 medium, 10% FBS, and 1% penicillin/streptomycin. The following day, the medium was replaced with a fresh medium, supplemented with either 5 nM (HCC827) or 25 nM (H1975 and PC9) osimertinib, which led to a significant loss in viability within 2 days. After 3–4 days of growth, surviving cells were trypsinized and plated again at the same initial density and the next day, the medium was replaced with a fresh growth medium, supplemented with 2× (relative to the previous concentration) of osimertinib. This process continued until osimertinib concentration reached 2 mM. The newly generated osimertinib-resistant cells (H1975-osiR-NCI1, PC9-osiR-NCI1, HCC827-osiR-NCI1, and HCC827-osiR-NCI2) were then maintained at 2 mM osimertinib.

### 4.4. Cell Extract and Mouse Tissue Extract Preparation, Immunoprecipitation, and Immunoblot Analysis

Cells were washed with ice-cold PBS and lysed in RIPA buffer (150 mmol/L NaCl, 1.0% IGEPAL CA-630, 0.5% sodium deoxycholate, 0.1% SDS, and 50 mmol/L Tris, pH 8.0) supplemented with protease and phosphatase inhibitors. Tissue extracts were prepared using a TissueLyser (Qiagen, Hilden, Germany) following the manufacturer’s protocol. Protein concentrations were quantified using the BCA method (ThermoFisher). Cell lysates were combined with 4× SDS sample buffer (Millipore Sigma, Burlington, MA, USA), incubated at 95 °C for 5 min and fractionated by 4–15% polyacrylamide-SDS PAGE. Proteins were transferred to nitrocellulose membranes using a semidry transfer method (Bio-Rad) and probed with the specific antibodies as described in the figures.

### 4.5. RNA POL II Ser 2 (pS2) Phosphorylation Detection by Western Blot

For pS2 detection by Western blot, Jurkat cells were seeded at a density of 1 × 10^6^ cells/mL in 5 mL of complete medium in a T25 flask. Cells were treated with the compounds for 6 h. After 6 h, cells were harvested and lysed with cell lysis buffer containing 50 mM HEPES (pH 7.4), 150 mM of sodium chloride, 5 mM of EDTA, and 1% of Triton ×100 (Sigma, Livonia, MI, USA), supplemented with protease and phosphatase inhibitor cocktails (Sigma, USA). Protein quantification was conducted using the BCA method (Thermo Scientific, Waltham, MA, USA, #23277). Total protein of 40 µg was resolved in 8% SDS-PAGE gel. Proteins were transferred onto PVDF-FL (Millipore # IPFL00010) membranes using the wet transfer method (30 V, 16 h). After protein transfer, PVDF-FL membranes were blocked using the Odyssey blocking buffer (Licor #927-50010) at room temperature for 1 h. pS2 RNA Pol II antibody at a dilution of 1:4000 (Bethyl # A300-654A) and β-actin primary antibody at a dilution of 1:20,000 (Santa Cruz #Ss-69879) were diluted in the Odyssey blocking buffer and added to the membrane and incubated over night at 4 °C on a rocker. The membrane was washed 3× times with TBST for 10 min each and incubated with Licor anti-rabbit (Licor #926-32211) and anti-mouse (Licor #926-68070) secondary antibodies at 1:10,000 dilution in the Odyssey blocking buffer at room temperature for 1 h. Membranes were washed 3× times with TBST and scanned using an LICOR Odyssey™ infrared scanner in the 800 and 680 channels. The raw data image file was stored in an LICOR proprietary format. The percent inhibition of pS2 in treated samples was calculated relative to the untreated controls. The data were analyzed using GraphPad Prism 6.0 (San Diego, CA, USA).

### 4.6. Cell Growth and Growth Inhibition Assays

Osimertinib-sensitive (H1975, PC9, and HCC827) or osimertinib-resistant (H1975-osiR-NCI1, PC9-osiR-NCI1, and HCC827-osiR-NCI1) cells (1 × 10^6^) were seeded into 10 cm Petri dishes and grown in RPMI-1640 medium supplemented with 10% FBS, 1% penicillin/streptomycin, and 2 mM osimertinib for 3–4 days. Cells were trypsinized, replated in 10 cm Petri dishes, and grown in RPMI-1640, 10% FBS and 1% penicillin/streptomycin but without osimertinib prior to the assay. Approximately 1500 cells/90 mL were plated in 96-well plates with 4 wells for each treatment condition. Plates were incubated at 37 °C. For the cell growth assay, one set of cells was treated each day for 5 days with cell titer glow reagent (Promega), and luciferase luminescence (indicator of cell viability) was measured according to the manufacturer’s protocol. For the growth inhibition end-point assay, cells in 96-well plates were treated with target drugs as follows. Drugs were serially diluted from 100 mM to 0.001 mM in RPMI-1640 medium containing 1% DMSO, and 10 mL was added to cells (90 mL) for a final concentration of 10 mM to 0.0001 mM. Cells were then allowed to grow for 72 h before the cell viability assay using cell titer glow reagent (Promega, Madison, WI, USA) according to manufacturer’s protocol. Data were plotted in GraphPad Prism to generate the growth inhibition curves and to determine the inhibitory concentrations of drug (IC50) for 50% growth inhibition.

### 4.7. General Method for Kinase Screening

Screening of the compounds in the in-house kinome panel was conducted using time-resolved (TR) measurements of fluorescence with fluorescence resolution energy transfer (TR-FRET) assay and Kinase Glo assay formats. Out of 26 kinases, 20 kinases (ALK, c-SRC, FLT3, FLT4, JAK2, KDR, ZAP70, TRKA, Aurora A, INSR, PDGFR-β, AXL, MUSK, EGFR, FAK, c-MET, FGFR1, RON, ACK1, TIE2) were performed in TR-FRET assay and 6 kinases (IRAK4, GLK, PI3Kγ, GSK3β, PKCα, SGK1) were performed in Kinase Glo assay format. All the kinase assays were performed at their respective ATP Km concentration. Screening of the compounds was conducted at 0.1 µM and 1 µM and the percent inhibition was calculated.

### 4.8. Jurkat Cell Viability Assay

For Jurkat cell viability assay, cells were seeded at a density of 10,000 cell/well in 90 µL of complete RPMI media in 96-well U-bottom plate (Corning #CLS 3799) and incubated at 37 °C in a CO_2_ incubator until compound addition. The compounds AU-15506 and AU-16770 were diluted using half-log dilutions in DMSO starting from 5 mM, which served as 500× stock. Serial dilution was performed 9 times. From the 500× DMSO stock, 2 µL was diluted into 98 µL of media to obtain intermediate dilution (10× stock of desired concentration). From the intermediate dilution, 10 µL was added to respective wells in triplicate. The control wells were treated with 0.2% DMSO. The assay plate was incubated for 72 h at 37 °C in a CO_2_ incubator. After 72 h of incubation, the assay plate was centrifuged at 1500 rpm for 5 min at room temperature and 50 μL of media was removed from each well with a multichannel pipette, followed by the addition of 50 μL of XTT (Thermo Fisher Scientific #X6473) working solution. The assay plates were incubated in a CO_2_ incubator at 37 °C until color development. The assay plates were read at 465 nM in a plate reader (Molecular Devices, Spectromax M3). Percent inhibition of proliferation of compound-treated wells was compared to that in DMSO-treated wells. The data were analyzed using GraphPad Prism 6.0 (San Diego, CA, USA).

### 4.9. CDK12 and CDK7 Target Engagement Assay

Jurkat cells were cultured in RPMI media (Lonza 12-115F) with 10% FBS. For the target engagement assay, cells were seeded at a density of 1 × 10^6^ cells/mL in 7 mL of complete medium in T25 flask. Cells were treated with the compounds for 6 h. After 6 h, cells were harvested and lysed with a cell lysis buffer containing 50 mM HEPES (pH 7.4), 150 mM of sodium chloride, 5 mM of EDTA, and 1% of Triton ×100 (Sigma, USA), supplemented with protease and phosphatase inhibitor cocktails (Sigma, USA). Protein quantification was conducted using the Pierce BCA assay kit (Thermo Scientific #23277) following the supplier’s protocol.

#### 4.9.1. Plate Based CDK12 Target Engagement

In 96-well plates (Sigma #CLS-9018), 200 µg protein sample for each treatment was added to 100 µL volume of 1× HEPES lysis buffer. A total of 100 µL lysis buffer was used as a blank. To each sample, 1 µL of 100 mM DTT and 1 µM of Bio-THZ531 in DMSO were added. The samples were transferred to a plate shaker and incubated overnight at 4 °C. On the following day, the samples were incubated in shaker at RT for 4 h. Precoated streptavidin microplates (Thermo Fisher #15501) were equilibrated at RT for 15 min and washed 4 times with 200 µL of reagent diluent each time (Tris-buffered saline with 0.05% tween and 0.5% BSA). A total of 100 μL of each of the Bio-THZ531 incubated samples was transferred to the streptavidin-coated microplate and incubated for 2 h at RT on a plate shaker. The sample from the microplate was discarded and the plate was washed 4 times with 200 µL of reagent diluent each time. CDK12 primary antibody (CST #11973s) was diluted in reagent dilution 1:1000, and 100 µL of the antibody dilution was added to each well in the microplate. The plate was incubated overnight at 4 °C on a plate shaker. The diluted antibody from the microplate was discarded and the plate was washed 4 times with 200 µL of reagent diluent each time. A total 100 μL/well of HRP-labelled anti-rabbit antibody (Invitrogen #656120) (1:3000 dilution in reagent diluent) was added to the microplate and incubated for 2 h at room temperature on a plate shaker. The diluted antibody from the microplate was discarded and the plate was washed 5 times with 200 µL of reagent diluent each time. A total of 100 μL of TMB substrate (ThermoFisher #34028) was added to each well in the microplate and incubated at RT. The development of blue color was monitored, and the assay was stopped by adding 50 µL of 2N Sulfuric acid in each well. Absorbance was measured at 450 nm and 570 nm wave lengths on a plate reader (Molecular Devices Spectromax M3). The percentage of CDK12 occupancy was calculated for compound treatment samples with reference to the untreated samples. The untreated sample was considered to have an occupancy of 100%. The data were analyzed using GraphPad Prism 6.0 (San Diego, CA, USA).

#### 4.9.2. Plate-Based CDK7 Target Engagement

A high-binding microplate (ThermoFisher #15501) was coated with 100 µL CDK7 antibody (Bethyl Labs #A300-405A) (1:500 dilution in PBS) in each well and incubated overnight at 4 °C. The following day, coating buffer was discarded, and the plate was washed 4 times with 300 µL reagent diluent (Tris-buffered saline with 0.05% tween and 0.5% BSA). A total of 200 µL of blocking buffer (Tris-buffered saline with 0.05% tween and 2.5% BSA) was added to the wells in the microplate and incubated at RT for 2 h on a plate shaker. The blocking solution was discarded, and the microplate was washed 4 times with 300 µL reagent diluent. A total of 200 µg of cell lysate from each treatment in 1× HEPES lysis buffer was added to the wells in the microplate. The microplate was incubated for 2 h at RT on a plate shaker. The samples in the plates were discarded and the microplate was washed 4 times with 300 µL reagent diluent. A total of 100 µL of 1 µM Bio-THZ1 was added to the wells in the microplate and incubated overnight at 4 °C. The next day, the microplate was incubated at RT for 4 h in a shaker. The Bio-THZ1 solution was discarded, and the microplate was washed 5 times with 300 µL reagent diluent. A total of 100 µL 1× streptavidin–HRP was added to the wells in the microplate and incubated at room temperature for 1 h on a plate shaker. After 1 h of incubation, the solution in the microplates was discarded and the wells were washed 7 times with 300 µL of reagent diluent and 100 µL of TMB substrate was added to the well and incubated at RT. The development of blue color was monitored, and the assay was stopped by adding 50 µL of 2N Sulfuric acid in each well. Absorbance was measured at 450 nm and 570 nm wave lengths on plate reader (Molecular Devices Spectromax M3). The percentage of CDK7 occupancy was calculated for compound treatment samples with reference to the untreated samples. The untreated sample was considered to have an occupancy of 100%. The data were analyzed using GraphPad Prism 6.0 (San Diego, CA, USA).

### 4.10. Cell-Cycle Analysis

PC9 and PC9-osiR-NCI1 cells were initially grown in an RPMI-1640 medium supplemented with 10% FBS, 1% penicillin/streptomycin, and 2 mM osimertinib. Following 3–4 days of growth, cells were trypsinized, replated in 10 cm Petri dishes and grown in drug-free RPMI-1640, 10% FBS, and 1% penicillin/streptomycin. Both groups of cells were treated with DMSO (vehicle), osimertinib (50 nM), or AU-16770 (100 nM) for 18 h. Cells were trypsinized, washed with PBS, fixed in 70% ethanol, washed twice in PBS, and approximately 5 × 10^5^ cells/mL were incubated in 50 mg/mL propidium iodide/PBS and 100 units/mL RNase to label the DNA for 1 h before it was analyzed by FACS.

### 4.11. Quantitative PCR

PC9-osiR cells were grown in 60 cm plates (in triplicate) in a drug-free medium as described above. Cells were then treated with either DMSO (vehicle) or AU-16770 for 0 h, 8 h, 24 h, and 48 h, then lysed in an RLT buffer (Qiagen) containing 2-Mercaptoethanol. Total RNA was then purified using the RNeasy Mini kit (Qiagen) following the manufacturer’s protocol. A high-capacity RNA to cDNA kit (ThermoFisher) was used to synthesize random primed cDNA from 1.5 ug DNAse-treated RNA. Real-time PCR was conducted in 384-well plates using a ViiA7 Real-time PCR system (Applied Biosystems, Waltham, MA, USA). Singleplex reactions (10 μL) containing an FAM-MGB expression assay for the gene of interest (ATM Hs01112326_m1, ATR Hs00992138_m1, BRCA1 Hs00212914_m1, BRCA2 Hs01037421_m1, CHEK1 Hs00967510_g1, FANCI Hs01105312_m1, MDC1 Hs01029034_m1, NEK9 Hs00929599_m1, ORC3 Hs01031861_g1, RAD51D Hs00979545_g1, TERF2 Hs01030573_m1) or GAPDH endogenous control (ThermoFisher) were performed using cDNA synthesized from 20 ng of RNA and 1× Universal Master Mix (ThermoFisher—without Amp Erase UNG). The comparative Ct method (delta, delta Ct) was used to determine the relative expression normalized to GAPDH (Applied Biosystems^®^ ViiA™ 7Real-Time PCR System Getting Started Guides). Samples were analyzed in quadruplicate, and values were expressed as the mean ± SE.

### 4.12. Copy Number Assay

Genomic DNA was isolated using the AllPrep DNA/RNA Mini kit (Qiagen) following the manufacturer’s protocol. Real-time PCR was conducted in 384-well plates using a ViiA7 Real-time PCR system (Applied Biosystems). Multiplex reactions (10 μL) containing an FAM-MGB copy number assay for ERBB2- (Hs00159103_cn) or MET (Hs01602615_cn) and a VIC TAMRA TERT copy number reference assay (ThermoFisher) were performed using 10 ng genomic DNA and 1x Universal Master Mix (ThermoFisher—without Amp Erase UNG). The comparative Ct method (delta, delta Ct) was used to determine the relative copy number normalized to TERT, relative to the parental cell line (Applied Biosystems^®^ ViiA™ 7Real-Time PCR System Getting Started Guides). Samples were analyzed in quadruplicate, and values were expressed as the mean ± SD.

### 4.13. Xenograft Studies

All mice xenograft studies were conducted on 5–6-week-old female athymic nude mice (NSG) with approval from the NCI Animal Care and Use Committee (ACUC). Mice were obtained from NCI-Frederick and maintained in a pathogen-free facility at the NCI. H1975-osiR-NCI1 or PC9-osiR-NCI1 cells were cultured in RPMI (Millipore-Sigma) supplemented with 10% fetal bovine serum (Gemini) and 1% penicillin/streptomycin. All cell lines were then cultured in a humidified incubator with 5% CO_2_ at 37 °C. Cells were detached using 0.25% trypsin (ThermoFisher) and resuspended in PBS prior to implantation. Approximately 2 × 10^6^ cells/0.1 mL/mouse were implanted subcutaneously. Between 7–10 days post-injection, mice were visually checked for pulpable tumor formation and treated (oral gavage) with 5 mg/day/kg body weight for another 12–14 days. Tumor growth was monitored twice weekly by bilateral caliper measurement and tumor volume was calculated using the length × diameter^2^ formula. Mice with tumor volume <200 mm^3^ were randomized into vehicle or treatment groups to ensure equal distribution across groups. However, for tumor tissue collection and lysate preparation, mice with xenografts were maintained with osimertinib until tumor size reached approximately 900 mm^3^ prior to drug treatment for 5 days. Vehicle (0.5% methyl cellulose), osimertinib (5 mg/kg/day), AU-15506 (5 mg/kg/day), or AU-16770 (5 mg/kg/day) were administered by oral gavage. AU-15506 or AU-16770 were prepared in a solution containing 10% PEG400, 10% D-ɑ-tocopheryl polyethylene glycol succinate (TPGS), and 0.4% Tween 80. Tumor size was measured twice weekly. Tumor volume for each mouse was converted to percent change based on a baseline volume of 100% on treatment day 0. Tumor growth inhibition was assessed by comparison of either the average tumor volume or mean changes in tumor volume for the control and treatment groups. Statistical significance was evaluated using a two-tailed Student’s *t* test.

### 4.14. In Vivo Preclinical Pharmacokinetic Analyses

Animal experimental procedures used in this study were approved by the Institutional Animal Ethical Committee based on the Committee for the Purpose of Control and Supervision on Experiments on Animals guidelines. Mice were used for the experiment after one week of acclimatization to standard laboratory conditions. Mice were fed with standard diet and water ad libitum.

Pharmacokinetic profiling was carried out using oral and intravenous route (IV) dosing. Three mice per route were administered with the test compound at indicated dose levels dissolved in specified vehicle (intravenous arm contains 2% *v*/*v* NMP + 30% *v*/*v* PEG + QS saline, oral arm contains 0.2% *w*/*v* tween 80 and 0.5% *w*/*v* methyl cellulose). Bolus IV dosing was performed in male CD1 mice, whereas oral dosing was conducted through oral gavage. The plasma samples were collected at designated time points and were frozen and stored at below −70 °C until analysis. The plasma samples were deproteinized with acetonitrile containing the internal standard, followed by centrifugation, and the supernatants were used for analysis. Quantitative bioanalysis of the AU-15506 and AU-16770 in the plasma samples was conducted using the fit-for-purpose LC-MS/MS method.

The PK data were expressed as mean ± standard deviation and PK parameters were determined using WinNonlin 8.1 software. The plasma concentration after injection (C0 min), the area under the concentration–time curve from time zero to 24 h (AUC0-t), Vdss, and CLtotal for after IV administration were obtained. The maximum plasma concentration (Cmax), time to maximum plasma concentration (Tmax), AUC0-t, and %F after oral administration were also obtained.

## 5. Conclusions

In conclusion, these CDK12/13 inhibitors may serve as a new generation of drugs for use as a combination therapy with osimertinib to circumvent resistance. In addition, this drug combination may be tested as a frontline therapy to potentially delay the development of osimertinib resistance and increase the durability of response to frontline therapy of EGFR mutant lung cancer that needs to be tested in pre-clinical models and subsequently in clinical studies.

## Figures and Tables

**Figure 1 cancers-15-02263-f001:**
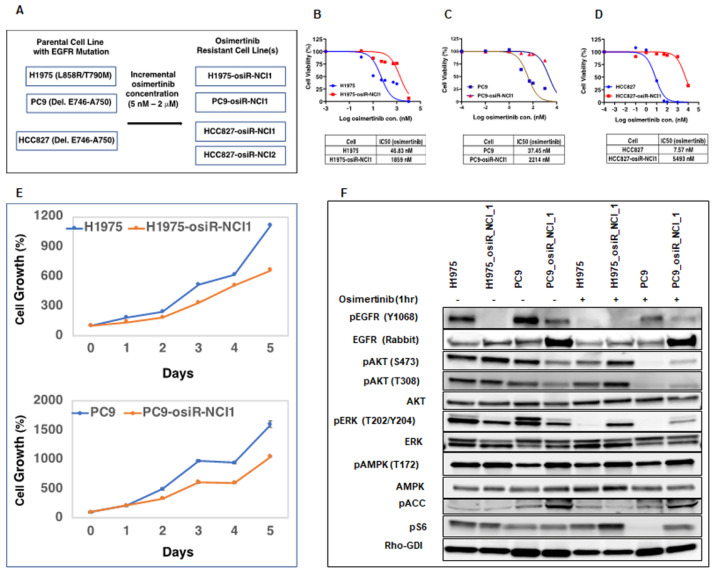
Generation of osimertinib-resistant lung adenocarcinoma cell lines. (**A**). Osimertinib-resistant isogenic H1975-osiR-NCI1, PC9-osiR-NCI1, HCC627-osiR-NCI1, and HCC627-osiR-NCI2 cell lines were generated from osimertinib-sensitive H1975, PC9, and HCC827 cells by osimertinib dose escalation. (**B**–**D**). Cell-growth-inhibition end-point assays were performed by growing osimertinib-sensitive and -resistant cells in drug-free medium for 3 days prior to transferring to medium containing serial dilutions of osimertinib. IC50s indicate that osimertinib-resistant cells have significantly higher drug tolerance (1859 nM–5493 nM) than osimertinib-sensitive cells (7.57 nM–46.83 nM). (**E**). Five-day cell growth assay indicates that osimertinib-resistant cells have a slower growth rate than osimertinib-sensitive cells. (**F**). Osimertinib-sensitive and -resistant cells were grown in drug-free medium for 3 days prior to treatment with DMSO (control) or 50 nM osimertinib for 1 h. Cell lysates were immunoblotted with indicated antibodies. Rho-GDI antibody was used as a loading control.

**Figure 2 cancers-15-02263-f002:**
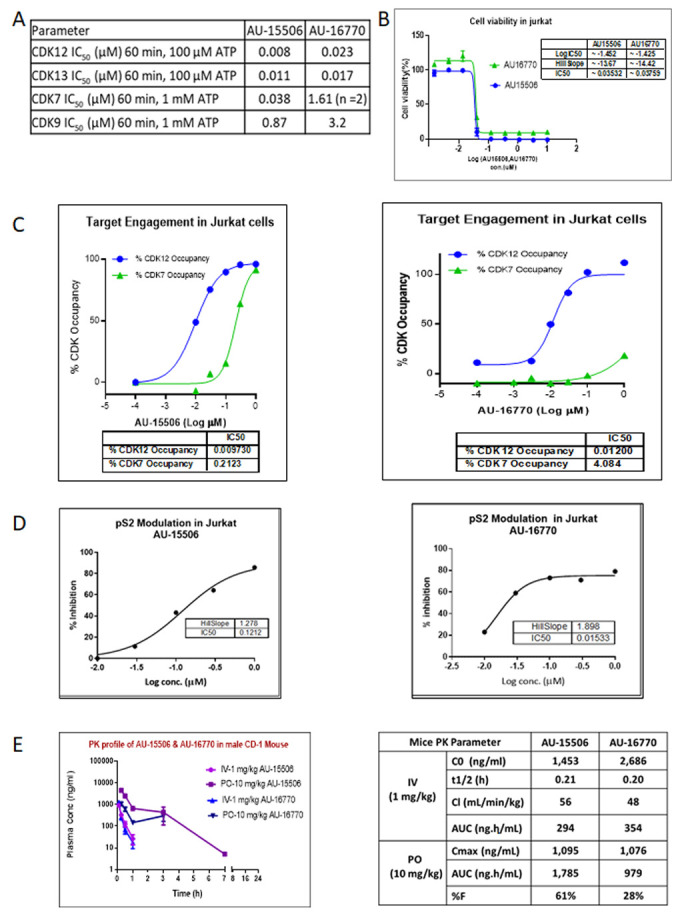
Characterization of lead compounds targeting CDKs. (**A**) In vitro profile of lead compounds AU-15506 and AU-16770. Kinase inhibition assay was performed with indicated kinases in presence of above CDK inhibitors. (**B**) Cell viability profile of lead compounds AU-15506 and AU-16770 in Jurkat cells. IC50s of these inhibitors are shown in the inset table. (**C**) Target engagement profile of lead compounds AU-15506 and AU-16770 in Jurkat cells. % CDK occupancy of the inhibitors increases with increasing concentration and IC50s of %CDK occupancy is shown in the bottom table. (**D**) Modulation of pS2 in Jurkat cells treated with lead compounds AU-15506 and AU-16770. (**E**) Pharmacokinetic profile of lead compounds. Mice were separately administered with AU-15506 and AU-16770 either by intravenous injection (IV) or by oral gavage (PO), and the pharmacokinetic parameters were measured at different time points as indicated. Abbreviations: C_0_, initial plasma concentration; T_1/2_, the mean terminal elimination half-time; CL, clearance; AUC, area under plasma concentration; C_max_, maximum plasma concentration; %F, bioavailability.

**Figure 3 cancers-15-02263-f003:**
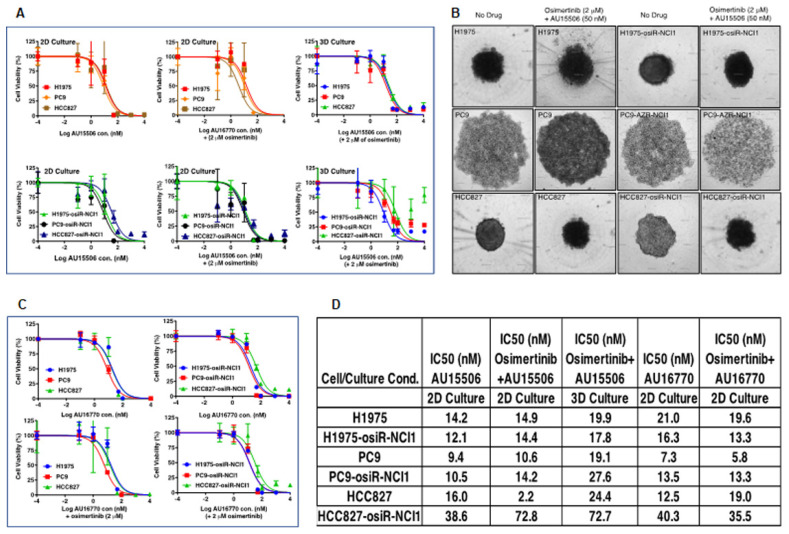
CDK12 inhibitors, AU-15506 and AU-16770, inhibit growth of osimertinib-sensitive and -resistant cell lines. H1975/H1975-osiR-NCI1, PC9/PC9-osiR-NCI1, and HCC827/HCC827-osiR-NCI1 were either grown in flat-bottom (2D culture, (**A**)-left and middle panels) or specially designed round-bottom (3D sphere culture, (**A**)-right panels, and (**B**)) 96-well plates, and were then treated with AU-15506, AU-15506-osimertinib (**A**,**B**), AU-16770, or AU-16770-osimertinib (**C**) as indicated for 3 days before cell viability measurements. AU-15506 or AU-16770 either alone or in combination with osimertinib inhibited cell growth as indicated by lower IC50s (**D**).

**Figure 4 cancers-15-02263-f004:**
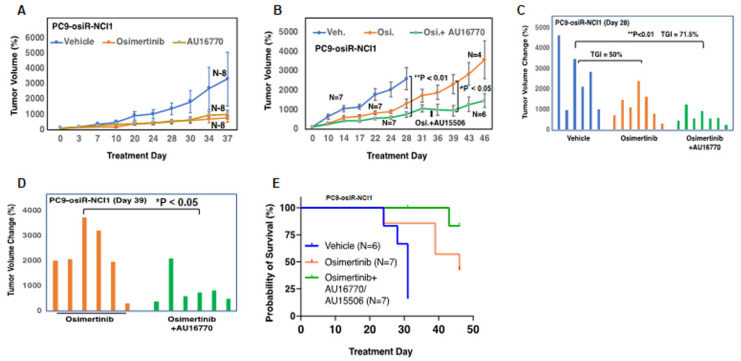
AU-16770 in combination with osimertinib inhibits tumor growth in PC9-osiR-NCI1 and H1975-osiR-NCI1 subcutaneous xenograft mouse models and increases overall survival. (**A**) Mice bearing PC9-osiR-NCI1-derived tumors were treated for 37 days with vehicle (N = 8), 5 mg/kg/day osimertinib (N = 8), or 5 mg/kg/day AU-16770 (N = 8). (**B**) Mice bearing PC9-osiR-NCI1-derived tumors were treated for 46 days with vehicle (N = 7), 5 mg/kg/day osimertinib (N = 7) or AU-16770-osimertinib (N = 7) (days 0–32), and AU-15506-osimertinib (days 33–46) in combination. Plot of daily percent tumor volume change shows AU-16770–osimertinib combination treatment reduces tumor size. (**C**,**D**) Bar plot of percent tumor volume changes in each mouse representing (**B**). Overall tumor burden is significantly reduced in mice treated with AU-16770–osimertinib and AU-15506–osimertinib combination treatment compared to vehicle (**C**) or osimertinib (**D**) Tumor growth inhibition (TGI) is calculated by [1 − mean tumor volume of treatment group/tumor volume of vehicle control group] × 100. (**E**) Kaplan–Meier survival curves of mice bearing PC9-osiR-NCI1-derived tumors treated with vehicle (N = 6), osimertinib (N = 7), or AU-16770–osimertinib and AU-15506–osimertinib (N = 7) combination.

**Figure 5 cancers-15-02263-f005:**
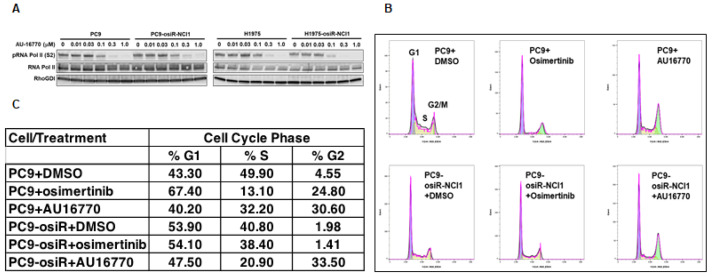
Cell-cycle analysis reveals AU-16770 treatment of PC9 and PC9-osiR-NCI1 cells causes G2 arrest. (**A**) PC9 and PC9-osiR-NCI1cells were treated with 0, 0.01, 0.03, 0.2, 0.3, and 1.0 μM AU-16770 for 3 h. Cell lysates were immunoblotted with RNA Pol II antibodies and Ser2 or-Ser5 phospho-specific RNA Pol II antibodies to determine the effective concentration of AU-16770 required to inhibit RNA Pol II-Ser2 phosphorylation, a marker of CDK12 activity. RhoGDI was used as a loading control. (**B**) Cell-cycle analysis of PC9 and PC9-osiR-NCI1 cells by flow cytometry. Cells were treated with DMSO (control), 50 nM osimertinib, or 100 nM AU-16770 and stained with propidium iodide. (**C**) Relative quantification of the proportion of cells in G1, S, and G2 phases shows that AU-16770, but not osimertinib treatment, causes a significant proportion of PC9-osiR-NCI1 cells to accumulate in G2.

**Figure 6 cancers-15-02263-f006:**
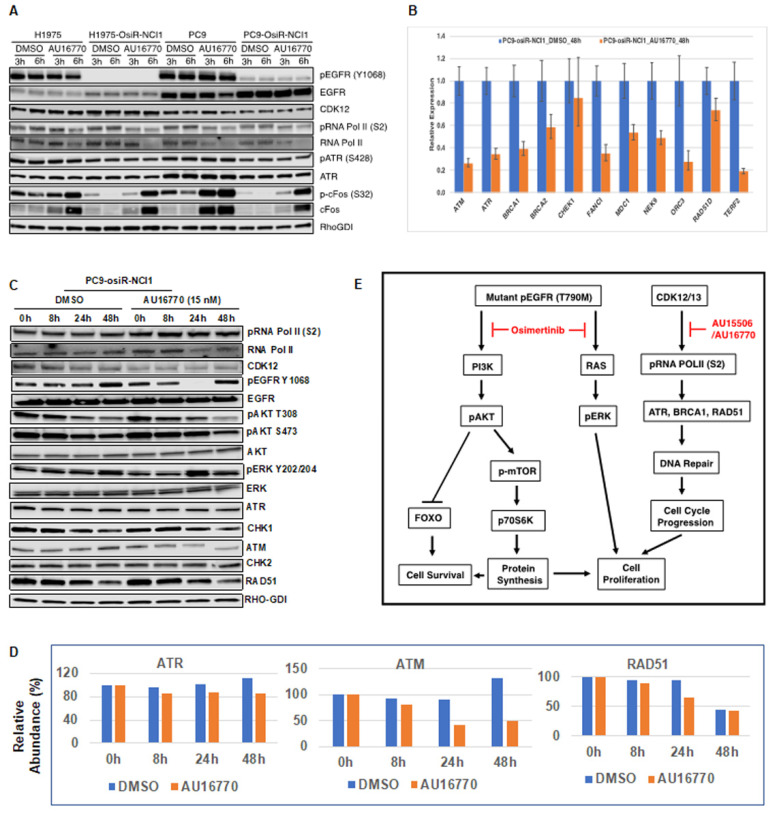
Identification of cellular components affected by AU-16770. (**A**) H1975/H1975-osiR-NCI1 and PC9/PC9-osiR-NCI1 cells were treated with 100 nM AU-16770 for three and six hours and then lysed in RIPA buffer. Cell extracts were immunoblotted with phospho-specific or total antibodies against EGFR, AKT, ERK, RNA pol II, ATR, FOS, and CDK12. RhoGDI-specific antibodies were used for loading control. (**B**) Expression analysis of genes having long transcripts. mRNA was isolated from PC9-osiR cells in triplicate following treatment with DMSO (vehicle) or AU-16770 for 24 h. Relative expression of ATR, ATM, BRCA1, BRCA2, FANC1, MDC1, NEK9, ORC3, RAD51, CHEK, and TERF2 genes were analyzed by quantitative PCR using GAPDH as a reference gene. (**C**) Immunoblot analysis of extracts from PC9-osiR cells treated with DMSO or AU-16770 for 0 h, 8 h, 24 h, and 48 h with antibodies against EGFR signaling components, cell-cycle proteins, and DNA damage repair pathway proteins as indicated. AU-16770 specifically affected AKT survival pathway and DNA repair mechanisms. (**D**) Relative intensities of ATR, ATM, and RAD51 in PC9-osiR-NCI1 cells quantified from Figure 6C using ImageJ software. Intensities of above proteins at 0 h of drug treatment were represented as 100%. (**E**) Model depicting the combinatorial effect of osimertinib and CDK12 inhibitor (AU-15506 or AU-16770) in suppressing DNA replication/transcription of proliferation/survival-specific genes leading to inhibition of cell-cycle progression and cell proliferation in osimertinib-resistant cells. Osimertinib potentially upregulates receptor tyrosine kinases (RTKs) in osimertinib-resistant cells, potentially leading to sustained pERK and pAKT activity which are key to cell survival and proliferation. Inhibition of the CDK12 pathway by AU-15506/AU-16770 may lead to reduction in cFOS, ATR, BRCA1, and RAD51 activities and subsequent inhibition of cell-cycle pathway and cell proliferation.

## Data Availability

Not applicable.

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
