# Peer review of "Novel CDK12/13 Inhibitors AU-15506 and AU-16770 Are Potent Anti-Cancer Agents in EGFR Mutant Lung Adenocarcinoma with and without Osimertinib Resistance"

_cancers, 2023, doi:10.3390/cancers15082263_

Round 1

Reviewer 1 Report (Previous Reviewer 1)

Revised areas clarify the manuscript further. 

Reviewer 2 Report (Previous Reviewer 2)

I judge this revised manuscript to be fine.

This manuscript is a resubmission of an earlier submission. The following is a list of the peer review reports and author responses from that submission.

Round 1

Reviewer 1 Report

It is a well designed study. Few minor concerns are:

1. Please discuss possible reasons of decreased Erk phosphorylation in resistant cells-might be linked to decreased cell proliferation.

2. Does the CDK12/13 inhibition prevent or delay development of resistance to osimertinib? Any scientific data on it?

3. Line 132 concentrations of the inhibitors were written as mM- should be micro molar. Likewise please correct uM as micro molar.

4.Line 193 the animal model was described as orthotopic-it is not (injections were not into the lung tissue directly).

5. Please describe any possible unwanted effects of combination treatment.

Reviewer 2 Report

The authors showed CDK12/13 inhibitors AU-15506 and AU-16770 are potent anti-cancer agents for osimertinib-acquired resistant EGFR-mutated NSCLC.

This idea is novel and interesting. However, there are several critical concerns.

Major comments

1.      CDK12/13 inhibitors were shown to be effective against both sensitive and resistant cells to osimertinib. Thus, CDK12/13 inhibitors appear to be drugs that inhibit the growth of all cells rather than drugs that overcome resistance to osimertinib. Therefore, it is difficult to understand why only cells that have become resistant to osimertinib are being targeted with CDK12/13 inhibitors in this study. From the beginning of treatment, the authors should compare the effect of osimertinib and osimertinib plus CDK12/13 inhibitors on the sensitive cells, and should investigate whether osimertinib plus CDK12/13 inhibitors can lead to cure of EGFR mutation-positive lung cancer and prevent the acquisition of resistance.

2.      Are other resistance mechanisms such as C797S, MET, and HER2 not detected in osimertinib-resistant cells? The authors should examine.

3.      Figure 3

Why is the IC50 higher with AU15506 + osimertinib than with AU15506 in HCC827-osiR-NCI1?

4.      Figure 4

Why is there a lack of effect of AU15506 in Figure 4, unlike the results in Figure 3?  Is the amount of AU15506 used in vivo smaller than in Figure 3?

5.      Figure 4

In Figures A-C, the authors should observe long enough as in Figure D-F.

6.      Figure 4H

The authors have to do the experiments as separate groups like Osimertinib + AU15506 and Osimertinib + AU16770. Since the authors state that AU15506 is invalid in the in vivo model.

7.      Figure5 and 6

The authors should also conduct experiments on AU15506. Currently, AU15506 has not been studied. The authors should not describe AU15506 as a drug that overcomes osimertinib resistance in the discussion section.

8.      Figure 6A

No difference in ATR protein expression between DMSO and AU16770 in H1975-OsiR-NCI1 and PC9-OsiR-NCI. This is not consistent with the results in Figure 6B.

9.      Figure6C

No difference in ATR, CHK1, ATM, CHK2, RAD51 protein expression between DMSO and AU16770 in PC9-OsiR-NCI. The authors should conduct the experiment again.

Minor points

1.      Figure 2

Isn't it the opposite where you state 96% and 28%?

2.      Figure 4G

Is it correct that the p value is < 0.5, not 0.05?
